# Design a New Type of Laser Cladding Nozzle and Thermal Fluid Solid Multi-Field Simulation Analysis

**DOI:** 10.3390/ma14185196

**Published:** 2021-09-10

**Authors:** Yuan Zhang, Yexin Jin, Yao Chen, Jianfeng Liu

**Affiliations:** Mechanical and Power Engineering School, Harbin University of Science and Technology, Harbin 150080, China; zhangyuan1966@163.com (Y.Z.); cy18746196627@163.com (Y.C.); a13654682680@163.com (J.L.)

**Keywords:** new laser cladding nozzle, multiple powder feeding channels, powder flows, new heat source, thermal deformation

## Abstract

Coaxial powder feeding technology in the field of metal additive manufacturing is booming. In this paper, a new laser cladding nozzle with powder feeding channels of inner and outer rings is designed. The nozzle works with a new kind of laser, which is a new heat source with an inner beam and outer beams. The water-cooling channels are simulated in Ansys Workbench. The simulation results present the temperature distribution of the working nozzle and the velocity of the cooling water. The thermal dilation of the nozzle in the working environment is also simulated. The results show that the loop water cooling channel could effectively reduce the high temperature of the nozzle down to about 200 °C. In addition, it could well restrain the thermal deformation of the nozzle lower to 0.35 mm. The equivalent stress of most parts is controlled under 360 MPa. Then, the powder flows of the inner and outer rings of the multiple powder feeding channels are simulated in Ansys Fluent. The convergence effect of the powder flow could be assumed and some significant parameters, such as the velocity, are acquired. The results present that these multiple powder feeding channels could realize the generation and removal of removable supports of workpieces with highly complex shapes and achieve a large processing range and good processing efficiency. The velocity of the powder flow at the outlet is elevated to about 5 mm/s. Then, the thermal cladding states under the new laser heat source of the powder are simulated in Workbench. The temperature of the melting process and the thermal deformation and the equivalent stress/strain of the additive parts are obtained in the emulation. The results emerge that the powder melting range and the ascending temperature of the melting pool are improved with this effect. The greatest temperature of the melting pool is about 2900 °C in the machining process, and the maximum thermal equivalent stress is 1.1407 × 10^10^ Pa.

## 1. Introduction

With the development of science and technology, additive manufacturing technology (AM, commonly known as 3D printing technology) has been increasingly integrated into production. AM increasingly abates the prominent disadvantages of traditional processing. AM has the advantages of a short production cycle, large design freedom, and easy processing of highly complicated workpieces. Lightweight structure design is adopted and higher strength requirements can be achieved. With the innovation and development of the technology, requirements for workpieces performance are improving [1,2,3,4,5].

As a main part of AM, metal additive manufacturing attracts many experts and scholars to conduct research. There are two main methods to preset the metal material powder; one is powder feeding (or wire feeding) and the other is a powder bed. The main advantage of a powder bed is its high processing efficiency, which can process multiple workpieces of the same type at one time. The advantage of powder feeding is that more complex parts can be processed, and that complex shape requirements for a single part can be well obtained [6,7,8,9,10].

At present, the existing coaxial powder feeding method has some obvious shortcomings in terms of the workpieces, with more stringent quality requirements. In the past, the processing time of the coaxial powder feeding method was long, complicated workpieces were difficult to process, and the workpiece strength was insufficient. Therefore, a new kind of cladding nozzle, which could achieve a high efficiency and high quality, and that could complete highly complicated machining requirements, is necessary. Simulations, which include the temperature distribution and thermal deformation of a new nozzle in the working environment, are important for nozzle design. The powder flow convergence is shown in the simulations; these results are very important to present the machining effect. The thermal parameters of the additive forming parts are also necessary to be simulated to get results of the temperature and equivalent stress [11,12,13,14,15].

In this research, we design some water cooling channels to get the best result for a low temperature for a new cladding nozzle under the high-temperature effect of a laser. Then, we simulate the cooling water’s flowing state in the channels and the thermal expansion of the nozzle. We then simulate the powder flow jetting from the nozzle to get the results of its speed and its convergence state. We further simulate the additive part under a new kind of laser and get results of the temperature distribution and the equivalent stress/strain. From the research, we could get the results for a cladding nozzle working state and the quality of the additive part under a new kind of laser. We could design a better machine from the results and avoid unnecessary waste.

## 2. Structure Analysis and Design of Laser Cladding Nozzle

### 2.1. Structure Design of New Type Laser Cladding Nozzle

The processing quality of the existing coaxial powder feeding nozzle exceed the traditional manufacturing, but there are many things that can be improved. At present, there are two main powder feeding methods; one is laser outside powder feeding, and the other is laser inside powder feeding. At present, the laser outside powder feeding mode is more common. Many universities and enterprises have carried out in-depth research and manufacturing production in this area. The laser inside powder feeding method is mainly used in the research [16,17,18,19,20]. The processing diagrams of the two coaxial powder feeding methods are shown in Figure 1a,b.

The disadvantages of laser outside powder feeding are as follows: it has a low efficiency, small spot diameter, and small heating area. The microstructure is not uniform. The disadvantages of laser inside powder feeding are that the ring laser beam processing is difficult, and the mechanical properties of parts are relatively simple. Therefore, in this paper we designed a new coaxial powder feeding laser cladding nozzle. It ensures machining efficiency and improves the machining accuracy [21,22,23,24,25].

The laser cladding nozzle designed in this paper is mainly composed of central laser beam channels and inner 6-way powder feeding channels and the outer 9-way powder feeding channels, as well as protective gas channels and water-cooling channels. The structure is shown in Figure 2a,b.

The power range of the central laser was 0.5–2.5 kW. In this paper, a new type of laser-focusing form, with one spot in the center and three spots on the outside, was applied. The diameter of each spot of the central laser beam was about 5 mm. In an ideal state, the central laser beam could cause the central spot to form separately, make several spots form at the same time, and it could meet processing requirements under different working conditions. Therefore, when it machines workpieces with a high forming accuracy, the center spot could be converged to form a small range of heat affected areas to ensure that the amount of beam with powder converging–melting–solidification forming in unit time is relatively small, and it could ensure the processing density and forming effect of the workpieces. When the requirements of forming accuracy are relatively low, but forming area and forming speed per unit time are required, the inner and outer laser spots need to be formed at the same time to enlarge the range of the heat affected area. Therefore, the forming area per unit time is larger and the processing speed and efficiency are improved [26,27,28,29,30].

The inner 6-way powder feeding channels could process the required molding powder, such as titanium alloy powder, aluminum alloy powder, nickel-base alloy powder, etc. The alloy powder passes through the powder feeding channels, continues to move like a cone after spraying, and then converges at the processing surface (point). Because of the different processing conditions, for processing with a high molding accuracy, the center spot should be well coupled to the powder of the inner ring. For high efficiency machining, the inner and outer beam spots should be well coupled to the inner ring powder. For processing, which requires a higher molding efficiency, it needs to cooperate with the outer 9-way powder feeding channels to meet requirements. Metal powder could be ejected at the outlet through the inner 6-way powder feeding channels at a certain angle, and the powder converges with the laser beam in the processing area to form a cladding layer. The powder spraying angle is designed to be 60° with a horizontal plane, and the diameter of powder feeding channels is 5 mm. When machining high-complexity workpieces, it is necessary to form a removable support. Then the removable metal powder, such as copper powder or aluminum powder, can be added to three of the inner 6-way powder feeding channels (separation angle of 120°). At the same time, the removable powder should be different from the molded powder; that is, part of the molded part should not be affected during the removal.

The outer 9-way powder feeding channels have radius of 5 mm. They can process all the required molding powder or add removable metal powder into the symmetrical powder feeding channels. With the increasing demand for complexity of workpieces, previous metal additive technology could not meet the shape requirements of workpiece structures. A titanium alloy powder or a support layer powder could reach the processing area through the outer 9-way powder feeding channels to meet these processing needs. The support layer powder could be a copper powder or aluminum powder, so that the titanium alloy workpieces would not be damaged when the complex support is removed. Therefore, the copper powder or aluminum powder used in this paper were used as the removable support part and the copper support or the aluminum support were removed by immersion vibration with phosphoric acid or ferric oxide to meet the processing requirements of many complex structures. The Fe ion on the surface of ferric chloride is +3 valent, which has a strong oxidation. It can react with copper to form +2 valent ferrous, and oxidize copper to Cu^2+^ to form copper chloride and ferrous chloride. The chemical equation is:(1)$${\mathrm{Cu}+2\mathrm{FeCl}}_{{}_{3}}{=2\mathrm{FeCl}}_{2}{+\mathrm{CuCl}}_{2}$$

The reaction phenomenon is as follows: copper dissolves and the solution changes from brownish yellow (FeCl_3_) to blue (CuCl_3_), and the light green of FeCl_3_ is covered. Phosphoric acid is not a weak acid, so aluminum can react with phosphoric acid. However, it can only stay in the product stage of aluminum dihydrogen phosphate. Because the primary ionization of phosphoric acid is large, the secondary and tertiary ionizations are basically small and the concentration of produced H^+^ is very small. The product, aluminum dihydrogen phosphate, is insoluble in water and does not react with aluminum. The reaction is slower at room temperature. Heating or increasing the concentration of phosphoric acid could significantly improve the reaction. Aluminum dihydrogen phosphate is a very soluble substance in water. Titanium alloy does not react with phosphoric acid or ferric chloride.

The protective gas channels could be controlled and protected using argon. The radius of the argon inlet is 5 mm and the inner ring is used to transport argon for a better transportation of the gas. There are many choices for shielding gases, including argon, helium, etc. A protective gas can, not only produce a certain control effect on powder convergence, but also protect the processing process and reduce the spatter of some semi-finished powder in the hot-melt state, which has an impact on the workpieces and the surrounding environment [31,32,33,34,35].

In the water-cooling part, liquid water with a temperature slightly greater than zero flows in the water-cooling channels inside the laser cladding nozzle to control the temperature of the nozzle and reduce its thermal effect in the machining process. Because the laser in the formation process has a great influence on the surrounding temperature, if there is no cooling treatment, the powder feeding channels and the powder outlet would be affected. At the same time, the cooling should be sufficient and complete as much as possible to minimize the unnecessary heat effect in a limited amount of time and space to ensure the accuracy of processing and forming quality. This could make use of the gravity factor to make water flow more fully into the channels.

The powder flow converges with the laser in the processing area, and the high temperature of the laser makes the powder melt rapidly, thus forming a cladding layer in the processing area. The thickness and width of the cladding layer are related to powder feeding efficiency, spot diameter, laser power, and scanning speed. After the heat source leaves the newly processed area, under the influence of the cooling environment, the cladding part solidifies rapidly to form the processing part. After continuous processing, each part of the workpieces is slowly processed.

### 2.2. Simulation Analysis of the Structure of Laser Cladding Nozzle

#### 2.2.1. Water Cooling Simulation

Cooling is very important for components with high-temperature heating. Effective water cooling can not only reduce the influence of high heat on the thermal stress of components, but can also improve processing efficiency and ability.
(2)$$T_{out}=T_{in}+\frac{Q}{\rho \cdot v\cdot C_{p}}$$
where $T_{out}$ is the outlet temperature of the cooling liquid; $T_{in}$ is the inlet temperature of the cooling liquid. In this paper, $T_{in}$ is about 35 K; Q is the total heat dissipation rate of the heating device; ρ is the density of the liquid; v is the flow rate of the cooling liquid; and $C_{p}$ is the specific heat capacity of the cooling liquid.

It is very important to calculate $T_{out}$, the maximum temperature of the cooling liquid outlet. If $T_{out}$ exceeds $T_{\max}$, a cooling effect cannot be achieved.

It is assumed that $T_{out}<T_{\max}$. The next step is to determine the normalized thermal resistance of cooling. The following equation is used:(3)$$\theta =(T_{\max}-T_{out})\cdot (A/Q)$$
where θ is the thermal resistance; $T_{\max}$ is the maximum temperature allowed for cooling; $T_{out}$ is the outlet temperature of the cooling liquid; A is the area of the cooling area; and Q is the total heat dissipation power of the heating device.

The simulation of loop water cooling was as follows:

We used Ansys Fluent software to do the simulations of the water cooling. Ansys Fluent software is an authoritative and powerful software for fluid simulations and thermodynamic simulations, and the results can be accurate and reasonable.

The temperature distribution around the laser beam is shown in Figure 3. The temperature radiation range of the central laser beam is obviously reduced. The temperature gradient also decreases quickly. It could be quickly reduced to about 600 K, which can reduce the internal temperature of the laser cladding nozzle more effectively. This is more conducive to the spray and convergence of powder, and has a more effective cooling effect on subsequent processing.

The temperature distribution of cooling water channels is shown in Figure 4. The temperature at the entrance is low and increases gradually in the downward water flow process. The temperature increases gradually in the annular cone cooling channels. The highest temperature is about 450 K. The temperature distribution shows that the temperature rise near the inlet and outlet are faster than far from the inlet and outlet. According to the analysis, the reason for this is that water with a lower temperature first enters the back of the ring cone, then rises, and then enters the outlet. Because the temperature at the upper end of the cone is low, the temperature rise of this part of the flow is slow. After mixing with other parts of the water, the temperature generally increases to a higher temperature.

#### 2.2.2. Simulation of Stress Effect

The different temperatures not only affect the powder feeding effect and the forming effect of the workpieces, but also affect the thermal stress and deformation of the laser cladding nozzle itself. Therefore, it is necessary to simulate the temperature and thermal stress of the melting nozzle when the laser beam produces a high-energy heat source during processing. The analysis of the effect of thermal stress is obtained.

The principle of thermal deformation is very complex. In practical applications, the thermal expansion parameters of solid materials are expressed by the measured thermal expansion coefficient [36,37,38,39,40]. The thermal expansion coefficient can be divided into an average thermal expansion coefficient and a thermal expansion rate. The average coefficient of thermal expansion could be defined as the mean value of the relative change of the length of the corresponding sample when the temperature changes by 1 °C between $t_{1}$ and $t_{2}$, $a_{m}$ expressed in $10^{-6}$/°C. The calculation formula is as follows:(4)$$a_{m}=(L_{2}-L_{1})/\left[L_{0}(t_{2}-t_{1})\right]=(\Delta L/L_{0})/\Delta t$$

The thermal expansion rate (also known as coefficient of thermal expansion) is defined as the corresponding linear thermal (t) expansion value when the temperature changes by 1 °C under temperature T; $a_{t}$ expressed in $10^{-6}$/°C, and the calculation formula is as follows:(5)$$a_{t}=(dL/dt)L_{i}$$
where $L_{0}$ (mm) is the length of the sample when the temperature is $t_{0}$; $L_{1}$ (mm) is the length of the sample when the temperature is $t_{1}$; $L_{2}$ (mm) is the length of the sample when the temperature is $t_{2}$; and ΔL (mm) is the length of the sample between the temperature $t_{1}$ and $t_{2}$.

We used Ansys Workbench software to simulate the thermodynamics of the nozzle with the water-cooling effect under the thermal effect of the laser. This software is excellent for simulation of structural deformations under the statics, dynamics, the thermodynamics and electromagnetic mechanics, and so on. The simulations on the effects of the water cooling were finished and the results of the simulations are as follows:

The temperature distribution is shown in Figure 5a,b. The temperature dropped from the bottom to above. The temperature of the bottom end of the laser cladding nozzle reached 260 °C. The temperature of the other parts was about 200 °C. The temperature was far lower than the cladding point of the copper alloy. In previous content, the temperature of the cooling water at the outlet was obtained. The thermal effect of the nozzle was simulated using the temperature and other temperature distribution data. The results showed that the temperature of the part near the cooling water experienced a significant decrease at the bottom end of the nozzle or in other parts. It could be shown that the cooling effect was still very obvious. It could be said that the cooling water had an obvious effect on temperature control and reduction.

The influence of the thermal stress on the deformation of the nozzle is shown in Figure 5c,d. It can be determined that the maximum deformation position of the nozzle without cooling water is mainly distributed at the bottom of the nozzle. The maximum value was about 1 mm. The deformation distribution was slow, and the deformation value near the top of the nozzle was also large, and was about 0.5 mm. The results show that the deformation of the nozzle with cooling water was obviously reduced. The maximum value was about 0.35 mm, the deformation rate was very fast, and the deformation at the entrance of the cooling channels was reduced to about 0.15 mm. It could be said that cooling water has a good inhibition on the thermal deformation of the nozzle.

The equivalent stress of the thermal influence on the nozzle is shown in Figure 5e,f. It can be seen that the equivalent stress decreases obviously in the part with the cooling water. When there is no water-cooling part, the maximum equivalent stress was about 1600 MPa, and the equivalent stress value of the main equivalent stress distribution was about 500 MPa. When there was water cooling, the maximum equivalent stress value was about 650 MPa. And it is about half of the previous. The equivalent stress of the main distribution was about 150 MPa, and the average of the rest is decreased significantly. It can be seen that water cooling has a significant decrease in the equivalent stress of the nozzle.

The equivalent elastic strain of thermal effect on the nozzle changes is shown in Figure 5g,h. It can be seen that the equivalent elastic strain of the equal effect with water cooling was quite low. Through simulations of water cooling and without water cooling, it can be found that the water cooling had a good control and showed a reduction on the effect of temperature control, deformation of thermal stress, and equivalent stress and strain of the thermal stress.

In this section, through analysis and comparison of the existing traditional coaxial powder feeding methods, we could determine their advantages and disadvantages, and then design a new laser cladding nozzle using a new type of laser as a heat source and the new nozzle has six inner ring and nine outer ring powder feeding channels. Through control of the heat source and the cooperation of the powder feeding channels, workpieces with a high complexity could be processed. This could also ensure the accuracy of the processing, improve the quality of processing, and improve the speed and efficiency of processing.

In this section, after designing a new type of laser cladding nozzle, we finished the simulations of the loop-type water cooling channels, and comparative analysis showed that the loop-type water cooling channels had an excellent cooling effect. Furthermore, this paper applies the design of loop-type water cooling channels. After the design of the water-cooling channels was completed, the laser cladding nozzle with cooling water was simulated, and the temperature, deformation, equivalent stress, and equivalent strain under the effect of thermal stress were analyzed. Through comparative analysis, we could know that the cooling water played an important role in the working state of the nozzle.

## 3. Powder Flow Simulation and Analysis of the New Laser Cladding Nozzle

### 3.1. Theoretical Basis of Fluid Mechanics

As a common and important material state in nature, fluids are also very common in metal additive manufacturing. For example, in the process of conveying and spraying metal powder flow, a metal powder can be regarded as gas–solid mixed fluid. In processing, the different characteristics have a great impact on the processing effect. Therefore, simulation and analysis of powder flow are very important.

As an important part of analysis, hydrodynamic analysis needs to transform a known physical model or an unknown physical model into a mathematical model before simulation. Different software is used to set different parameters, according to different steps, to realize the simulation and verification of a physical model. In the analysis process of fluid mechanics, the fluid part to be studied should be regarded as a continuous medium to analyze and deduce the physical properties of the continuous medium [41,42,43,44,45].

In the range of the continuum model, matter is considered to be continuously distributed in the whole space. The physical parameters of the motion of matter are regarded as continuous and differentiable functions of space and time. According to the hypothesis of the model, the physical properties of a fluid medium, such as density, velocity, and pressure, could be regarded as continuous functions of space. The condition of a continuum model is that the average free path of a fluid is far less than the characteristic size of an object. There are several important parameters of continuous media and their formulas are shown in Table 1.

The powder particles in continuous medium are described by arbitrary Lagrangian Eulerian method:(6)$$\frac{d(m_{p}\nu )}{dt}=F_{t}$$
where $m_{p}$ is the mass of powder particles; ν is the movement speed of powder particles; and $F_{t}$ is the external force on powder particles.

The powder particles are affected by drag force $F_{y}$, exerted by the kinetic energy generated by the gas flow, and the gravity (g) of the Earth, exerted by the powder particles themselves, and has a downward movement trend. The formulas are as follow:(7)$$F_{t}=F_{y}+G$$
(8)$$F_{y}=\frac{1}{\tau_{p}}m_{p}(u^{t}-v)$$
(9)τp=4ρpdp23μCDRer
where $C_{D}$ is the drag coefficient and Rer is the relative Reynolds number of powder particles.

The formula of the time from the powder entering the laser beam and falling into the molten pool of the matrix can be seen as:(10)$$\Delta t=\frac{\rho_{p}Sr_{b}}{\phi_{m}\cos\theta }$$
where $\rho_{P}$ is the density of powder particles, the number is about 2500 kg/m^3^; $S_{rb}$ is the cross-sectional area of powder feeding port, the number is about 7.85 × 10^−4^; $\phi_{m}$ is the powder feeding rate, the number is about 1.75 m/s; and θ is the divergence angle of powder feeding port, the number is about 60°.

### 3.2. Simulation and Analysis of Powder Flow Convergence Process

Preprocessing is the basis of a good simulation effect. A good preprocessing could make the follow-up simulation process clear and concise. In this paper, SolidWorks software was used to establish a three-dimensional model in the design phase. In the preprocessing stage, we only needed to import the 3D model into Ansys Fluent software; however, it should be emphasized that, due to the need for simulation, the original three-dimensional model of the laser cladding nozzle could not be directly applied to the subsequent powder flow simulations. Because the powder spraying process needs to be simulated in the process of a simulation, it is necessary to draw part of a thin boundary on the basis of previous modeling to apply it in simulations of the spraying process.

It is shown in Figure 6 that after importing the supplementary 3D model, the powder feeding channels and boundary need to be filled to generate the simulation entity. In the simulation process, the simulation process and results of the powder flow in the powder feeding channels and the spray range are mainly observed, so the remaining entities could be compressed. The simulation process of the inner and outer loops is more consistent with the real machining state. In this paper, the range of powder spraying is estimated and the simulation boundary beyond the powder spraying outlet is drawn. The center laser beam outlet and the three outer laser beam outlets are then filled to prevent them from participating in the simulation process. The modeling results are shown in Figure 6. At the end of the early modeling, the model is meshed.

The analysis cloud chart shown in Figure 7 is the velocity cloud map in the ZX direction. This paper shows only a few powder feeding channels. The other channels are similar to these, so they are not covered. Through analysis of the simulation results, it could be found that the powder is a typical cone after ejecting. The powder form a certain convergence range in the corresponding processing area and the convergence range requires that the convergence diameter be larger than the diameter of the spot; there is no gap in the convergence center. The simulation results show that the powder flow in this work meets processing requirements. Through the color difference of the velocity cloud map, we could find that the velocities of different positions present different distributions. The velocity of the powder flow in the lower part of the powder feeding channels is larger. The maximum speed is about 5 m/s, mainly concentrated at the lower end and at the outlet of the powder feeding pipe. The maximum velocity distribution area is also at the outlet, which is conducive to powder convergence. The convergence of the powder has a very important impact on processing; if the convergence of powder is more concentrated, it could improve the processing efficiency and powder utilization. If the convergence of the powder is divergent, the processing efficiency may be low. Thus, the diameter of the inlet is designed to be smaller than the outlet. At the same time, if the outlet speed is reduced to about 1.5 m/s (on the premise of meeting the processing requirements), the pressure input of the powder feeding channels could be reduced to effectively reduce the energy consumption and cost. Through simulation of cooperative work, we could clearly reflect the actual situation in the work. Through analysis, it could be known that the convergence effect of end powder flow has been significantly improved. This not only enhances the convergence effect of the powder flow, but also makes the area of the convergence effectively fuse with the spot of the laser beam to realize the cladding of laser and the powder. The speed gradually decreases in the process of convergence, but is relatively uniform. It is analyzed that the jet pressure of the powder feeding channels is set because the acceleration of the gravity is set (during the process of processing). Thus, the total acceleration is a linear change in value. Therefore, the velocity decreases in a uniform gradient during the convergence process. Of course, there are also impact factors, such as the collision of powder particles during the process of aggregation. It could be found that the particle flow is columnar in a range of motion after aggregation, because the particles are free falling in the column’s shape. Gravity and the collision force of powder particles are the main factors.

It is shown that the velocity of the powder flow changes smoothly (showing a significant downward trend) in Figure 8. After 450 steps, the powder particles left the powder feeding channels and began to enter the convergence area. The velocity in different directions had different trends. The trend of the change in the early stage was the same. After entering the convergence area, the velocity in the X direction was stable and average. The velocity changes in Y direction and Z direction were following the trend of a steady oscillation. The velocity values were similar and were all less than the velocity values in the X direction. This is because the velocity in the X direction was not affected by many factors in the process of convergence. However, in other directions, it was influenced by forces (gravity, collision force, etc.). After 600 steps, the velocity parameters tended to be stable.

In this section, through simulation of the inner and outer rings working together, we could determine that the convergence effect of the powder end flow was very good, and the convergence angle was also very reasonable. In the injection range, the center single spot was fully surrounded, so that the heat energy generated by the light could be fully utilized. At the same time, its convergence range was relatively small, which could meet the requirements of processing small workpieces in terms of quality and precision. The powder feeding range, powder feeding speed, powder feeding angle, and powder utilization rate were greatly improved. In this way, when processing large-scale workpieces, they could meet short time requirements for large-scale workpieces. In this working state, the new laser spot could be fully coupled with light and powder, and the heat energy could be fully utilized. It can, not only ensure high machining quality and precision, but also greatly improve machining speed and efficiency.

## 4. Simulation and Analysis of Thermo Solid Coupling of New Laser Cladding Nozzle

### 4.1. Theoretical Analysis and Equation Establishment of Molten Pool Formation Theory and Heat Source

In the process of metal additive manufacturing, the heat source is a key factor. In the process of hot forming, metal is affected by the heat source. The powder will go through a process of “convergence cladding”. In the process of metal powder cladding, a small range of molten pool is formed. There are many physical phases in the molten pool and include molten liquid metal, solid–liquid mixing zone, and metal powder just entering the edge of molten pool. There are also many physical changes that occur in a molten pool. Such as transformations from solid to liquid, solid to gas and liquid to gas. In the process of a high-energy laser beam impacting a molten pool, a gasification plume can be formed due to the effect of recoil pressure. The vaporization plume may cause some objects in the molten pool to splash. In the process of molten pool flow, due to the influence of surface tension (the Marangoni effect), a series of dynamic changes can occur in the molten pool. Therefore, the simulation of a molten pool is key to the material forming process [46,47,48,49,50,51].

There are three ways to simulate the hot forming process in the field of metal additive manufacturing, these are shown in Table 2.

Different thermal analysis models can produce different simulation effects on the thermal effect process. With comprehensive consideration, this paper adopts the “heat conduction model based on continuum hypothesis”.

In the process of heat conduction, the heat source is also an important factor. The choice of heat source has different effects on the effect, time, and accuracy of a numerical simulation. There are three kinds of heat source models, shown below:
(1)Point concentration energy density model;(2)Volume heat source models (three);
(a)Gaussian body heat source model;(b)Double ellipsoid heat source model;(c)Gaussian heat source model based on high fidelity absorptivity;(3)Ray tracing heat source model.

The solidification rate is related to the laser scanning speed and the specific relationship is shown below:(11)R=Vcosθ
where R is the velocity at the front of solidification interface and its direction is normal along the solid–liquid interface; V is the scanning speed of the laser, the number is about 8 mm/s. θ is the angle between the normal direction of the solid–liquid interface and the scanning speed of the laser, the number is about 45°.

According to the formula, the solidification velocity on the surface of the molten pool is the highest, which is close to the scanning velocity (V), while the solidification velocity at the bottom of the molten pool is the lowest and is close to 0.

In addition, under the condition of rapid solidification, the thermal diffusivity in the solidification process is the main factor affecting the solidification rate of a metal.
(12)$$R\;(z,\tau )\;=\frac{-\sqrt{\alpha /\pi \tau }\cdot \exp[-{\left(\frac{z}{4\alpha \tau }\right)}^{2}]-\sqrt{\frac{\alpha }{\pi (t-\tau )}}\cdot \exp[-{\left(\frac{z}{4\alpha (t-\tau )}\right)}^{2}}{erf(\frac{z}{4\alpha \tau })-erf(\frac{z}{\sqrt{4\alpha (t-\tau )}})}$$
where t is the laser’s dynamic loading time, τ is the laser’s dynamic solidification time, and α is the thermal diffusivity in the solidification process.

It can be seen from the formula that the solidification rate (*r*) increases with an increase in thermal diffusivity α.

At the same time, the heat source model could be simplified. Assuming that the energy of laser heat source obeys a Gaussian distribution, the energy density is expressed as:(13)$$q(x,y)=\frac{2AP}{\pi \omega^{2}}\exp(-2\frac{r^{2}}{\omega^{2}})$$

The forming process satisfies the three-dimensional linear transient heat conduction control equation:(14)$$\rho c\frac{\partial T}{\partial t}(k\frac{\partial T}{\partial x})+\frac{\partial }{\partial y}(k\frac{\partial T}{\partial y})+\frac{\partial }{\partial z}(k\frac{\partial T}{\partial z})+Q$$
where ρ is the density of the powder, *c* is the specific heat capacity, *T* is the temperature, *t* is the interaction time between the laser heat source and the powder, *k* is the thermal conductivity, and Q is the heat input.

### 4.2. Simulation Analysis and Stress Analysis of Hot Cladding Process

In this paper, the simulation of transient heat is realized using Ansys Workbench for cladding of the central laser beam and the stress–strain simulation after the hot cladding, to better adjust the parameters and simulate more real processing conditions.

The moving heat source is a way to accurately simulate the thermal effect process of the laser heating process. In the actual processing process, the laser beam would converge with the powder flow at each processing point according to the planned path; therefore, under the simulation of a moving heat source, the heat effect data could be obtained to improve the convergence point and the radius of the powder flow. It would also make it more efficient with the laser beam cladding to better cooperate with each other [52,53,54,55,56].

The classical setting of a moving heat source is to use APDL to define the function, so the newly defined function and trajectory data and system data are used for thermal simulations, and then the parameters of the data are observed and some parameters of the thermal effects are obtained.

When the heat calculation is carried out by moving heat source, the initial equation of the heat source is defined in APDL, which can read the function. Then, the data of heat source motion trajectory is acquired, it can be applied to the command, and it can be added to the command of heat source call. This would specify the position of the heat movement to realize transient thermal simulations.

#### 4.2.1. Transient Thermal Simulation

(1)Parameter setting and heat source equation establishment

Because of the multi laser beam working process, multiple heat sources move and generate heat energy at the same time; therefore, the heat source equation is more complex than that of a single laser. Therefore, the movement of each flare needs to be set. In this way, the thermal effect could be ensured by multiple facula. A base square plate of 30 × 30 × 6 (mm) was used for simulations. The size of the square plate was not very large, because it was only a bearing entity in the simulation process of this work. The small square size moved is 2.2 × 2 × 0.6 (mm). Larger base sizes and more heat source fit sizes could be used. Unlike single laser beam simulations, the small square should be set in the range of each spot’s movement. In this way, the simulation and analysis of the solid thermal coupling caused by the subsequent thermal effects could be ensured. The size of each small square was the same; however, because the initial position and moving position of each spot were different, the position of each row of small squares was different. Due to the influence of the calculation time, this paper only simulates five mobile modules. The method of a life and death unit was adopted for the setup. It activate some unit module according to time plan to produce the effect of movement and generation. It is difficult to achieve a simulation of a moving heat source if the time over which the heat source is moving is consistent, or very similar to, the time generated by the moving block. This would have an impact on the subsequent mechanical characteristic simulations. Therefore, the time setting should be accurate.

Equation is:2.5 × 10^8^ × exp(((({x} − 0.011 − 0.008 × {time})^2^ + ({y} − 0.0011)^2)/0.0011^2^)

The significance of the parameters is as follows:

Thermal parameter (2.5 × 10^8^) is the value that determines the heating speed and speed of the heat source. The value is larger and the temperature increase is faster, which influences the temperature in a stronger way. The radius of the heat source was 1.1 mm, and the radius of the heat source refers to the diameter of spot in the simulation process. The setting of the value should be determined according to experience and the actual situation. Because the actual processing was considered when setting the moving block, the diameter of the heat source was set to the edge length of the moving block. The results show that the heat source is 11 mm away from the X axis, so that the simulation effect could be more obvious. Thus, the starting position of the simulation process was not at the edge of the base square plate, but in the middle position near the side. The change rate is 8 mm/s, so that different values could be set for the speed control survey to see what different values would affect the simulation results. The APDL is:

*DEL, FNCNAME……SF, A1, HFLUX, %BQ%

Equation is:2.5 × 10^8^ × exp(((({x} − 0.011 − 0.008 × {time})^2^ + ({y} + 0.0011)^2^)/0.0011^2^)

The significance of the parameters is as follows:

The thermal parameters were 2.5 × 10^8^. The radius of the heat source was 11 mm. The *x*-axis was 11 mm. The *y*-axis is shifted down to 11 mm and the change rate was 8 mm/s. The specific meaning was discussed above, so there is no more redundancy here. The same is true below. The APDL is:

*DEL, FNCNAME……SF, A2, HFLUX, %BW%

Equation is set as follows:2.5 × 10^8^ × exp(−3×(({X} − 0.009 − 0.008×{TIME})^2^ + ({Y})^2^)/0.0011^2^)

The significance of the parameters is as follows:

The thermal parameter was 2.5 × 10^8^. The radius of heat source was 11 mm. The *x*-axis was offset by 9 mm. The change rate was 8 mm/s and the APDL is:

*DEL, FNCNAME……SF, A3, HFLUX, %BR%

Equation is set as follows:2.5 × 10^8^ × exp(−3*(({X} − 0.01 − 0.008×{TIME})^2^ + ({Y})^2^)/0.0011^2^)

The significance of the parameters is as follows:

The heat parameter was 2.5 × 10^8^. The heat source radius was 11 mm. The *x*-axis offset was 10 mm. The change rate was 8 mm/s and the APDL is:

*DEL, FNCNAME……SF, A4, HFLUX, %BE%

(2)Analysis of simulation results

The results of simulation are shown in Figure 9. From the simulation results, the maximum temperature could reach 2900 °C when multiple laser beams worked at the same time. It was much higher than a single laser spot. At the same time, it could be clearly observed that the coverage of the high temperature was greatly improved. It was not only because of the linear increase in the number of light spots, but also because of the nonlinear enhancement when multiple light spots worked together. We took titanium alloy as an example; the cladding point of titanium alloy is about 1700 °C, and a range of temperatures higher than 1700 °C could cover almost all of the upper surface of the small squares in the heat source processing because the cladding point of titanium alloy is much higher than aluminum alloy, copper alloy, magnesium alloy, and other alloys. Therefore, this temperature effect could be applied to most alloys and the speed of the temperature drop is more gradual, which is more conducive to the processing of high-quality workpieces. Because it is a numerical simulation process, it could not be the same as a continuous and stable high temperature in an actual machining process; therefore, we only need to look at the simulation results in the high temperature band. Analysis showed that the heat source state was stable, and the moving speed was the specified speed, which was the planned laser beam moving speed. Many desired simulation results could be determined in the thermal analysis process, and subsequent simulations or experiments could be better optimized and improved through these results. In this paper, we couple the forming part of the “thermal solid” basing on the thermal analysis. Therefore, the mechanical properties of the workpieces in the process of hot forming could be analyzed. The mechanical analysis was related to the previous thermal analysis; therefore, five elements were still used to simulate the cladding and solidification to analyze the simulation results of the structural deformation.

#### 4.2.2. Simulation of Mechanical Effect of Thermal Stress

(1)Equivalent stress analysis

When an object is deformed due to external factors (stress, humidity, temperature field changes, etc.), the internal forces of the internal parts of the object interact with each other, and the internal forces per unit area are called stresses. Because of the existence of stress, it would induce stress to release and crack at the residual stress position. At the same time, due to the release process of residual stress in a short time at a high temperature and due to the local strength difference, the product would produce warpage or deformation at the residual stress position. Therefore, it is very important to study the stress of metal after cladding and solidification.

The results are shown in Figure 10. The simulation results show that the maximum stress is 1.1407 × 10^10^ Pa. However, the data are shown in the parameter data table in the process of simulation. Figure 10 shows the results at the end of machining; that is, the center of the heat source. Starting from this position, the stress changes smoothly in the direction of heat source movement, and the range of change is very small. It shows that the center of the moving direction of the heat source is always the place with the largest residual internal stress. Along the Y direction of both sides of the edge, the value of internal stress decreases smoothly. The minimum value (at the edge) was about 3.5 × 10^5^ Pa.

(2)Equivalent strain analysis

Engineering materials would produce strain under the action of stress. When the stress is small, there would be elastic strain. When the stress increases to a certain value, the relationship between the stress and strain is no longer proportional. When the stress disappears, permanent deformation would remain, which is called plastic strain. When a body is deformed by force, the degree of deformation at each point in the body is generally different. The mechanical quantity used to describe the degree of deformation at a point is the strain at that point. When plastic strain occurs in metal, it is accompanied by strain hardening. Due to the influence of thermal deformation, the metal materials in this work experienced the process of “cladding solidification”, so the plastic strain could be understood and analyzed temporarily.

It is shown in Figure 11 that the maximum values of equivalent strain decreased gradually from the center to the periphery. Of course, due to the increase in the number of laser spots, the influence range of the strain was obviously improved compared to that of single laser; however, due to its coupling relationship, the influence range of strain could produce a nonlinear and more stable decline. At the same time, it could be observed that the influence range of the strain is not only the range of the small squares, which is the part of the structure, but also has a certain impact on the substrate. In actual processing, the powder would penetrate into the substrate under the effect of high temperature heat energy, which has a certain impact on the substrate.

This section shows the simulations of thermal stress and some other results. In the simulation process, some parameters should be set, such as heat source function, heat source moving position code generation, and generation part of the settings. After the basic parameters are set, the simulation should be carried out. A series of results, such as the maximum temperature, the range of thermal influence, the trend of thermal change, the value and distribution of equivalent stress, and strain caused by thermal stress, could be obtained from the results. Therefore, it could be clearly determined that the working condition of multi laser beams and multi spots would have a good processing speed and efficiency. The forming quality and mechanical properties of the workpieces are well predicted and simulated.

## 5. Conclusions

In this work, a new kind of cladding nozzle is designed. The nozzle has six inner powder feeding channels and nine outer powder feeding channels. The channels could spray metal powder for forming machining and could also spray some removable metal powder for more complex processing. The nozzle uses a new kind of laser with one beam inside and three beams outside as a heat source. The new laser could provide a high temperature increase speed and a thermal impact range. The water-cooling channel is designed as a loop channel that can control the most of the temperature of the nozzle in a working environment down to about 200 °C. It is an appropriate temperature for the working nozzle. The thermal deformation of the nozzle is lower than 0.35 mm. The equivalent stress of most parts is under 360 MPa. These parameters are within the allowable range. Thus, the design of the new cladding nozzle is reliable.

Furthermore, the convergence of the powder flow is reasonable. It reflects a general range of powder spraying. After the improvement, the velocity of the powder flow at the outlet is about 5 mm/s. The velocity distribution is rational and the change is smooth and steady.

Furthermore, the greatest temperature of the melting pool is about 2900 °C in the machining process. It could fully melt metal efficiently. The maximum thermal equivalent stress is 1.1407 × 10^10^ Pa. The cladding states are shown in the figures. The results reflect that a great metal additive part could be obtained in the machining process.

As a result, we could get the advantages of the new kind of cladding nozzle from this research. Firstly, the new nozzle has inner 6-way powder feeding channels and outer 9-way powder feeding channels to meet the requirements of different working conditions. It could finish high-precision requirements at a normal speed and finish normal-precision requirements at a high speed. However, the previous nozzle could only meet a single working condition with low speed. Secondly, the new cladding nozzle could use different powders to machine the workpieces with highly complex shapes and remove the useless part with little loss of the used part. However, the previous nozzle could only machine workpieces of normal shapes. Thirdly, the new kind of cladding nozzle uses a new kind of laser to finish the powder heating and melting, and it could melt more fully and quickly than previous nozzles with other lasers; thus, the new kind of cladding nozzle could machine high-quality and highly complexity workpieces at a higher speed than previous nozzles.

## Figures and Tables

**Figure 1 materials-14-05196-f001:**
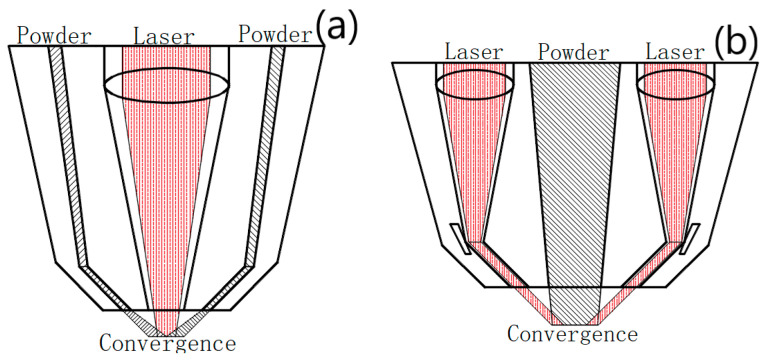
Schematic diagram of typical two feeding methods. (**a**) Laser outside powder feeding; (**b**) laser inside powder feeding.

**Figure 2 materials-14-05196-f002:**
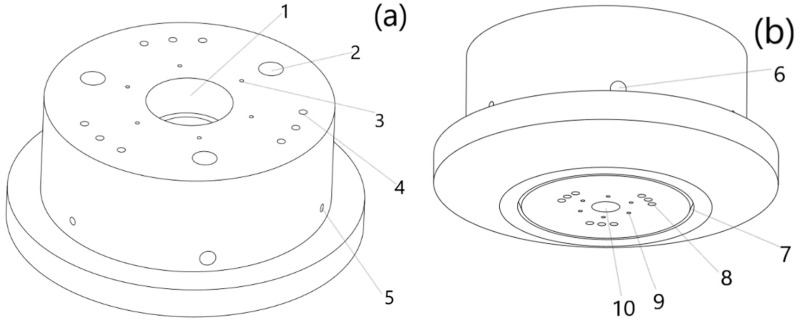
The structure diagram of the new cladding nozzle. (**a**) Vertical view; (**b**) bottom view. 1. Central laser beam inlet; 2. water cooling channels outlet; 3. inner 6-way powder feeding channels inlet; 4. outer 9-way powder feeding channels inlet; 5. protective gas channels inlet; 6. water cooling channels inlet; 7. protective gas channels outlet; 8. outer 9-way powder feeding channels outlet; 9. inner 6-way powder feeding channels outlet; 10. central laser beam outlet.

**Figure 3 materials-14-05196-f003:**
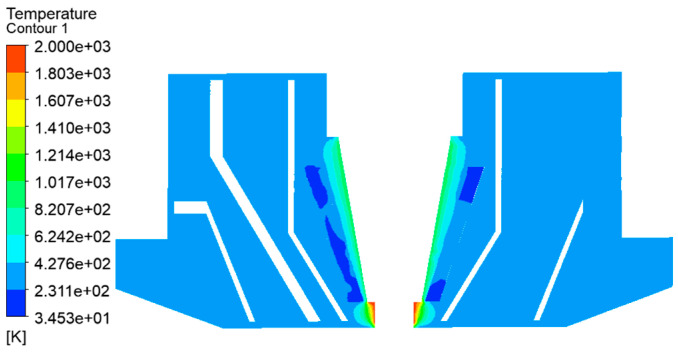
Internal temperature distribution.

**Figure 4 materials-14-05196-f004:**
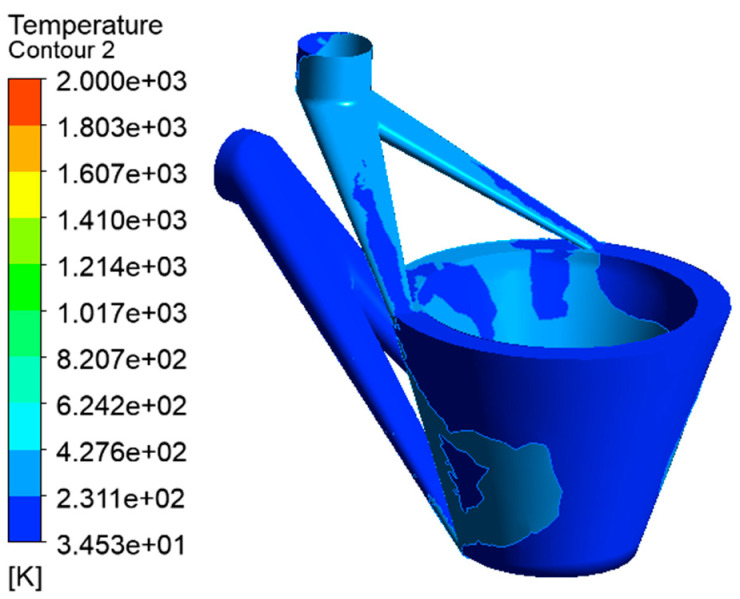
Temperature distribution of cooling water.

**Figure 5 materials-14-05196-f005:**
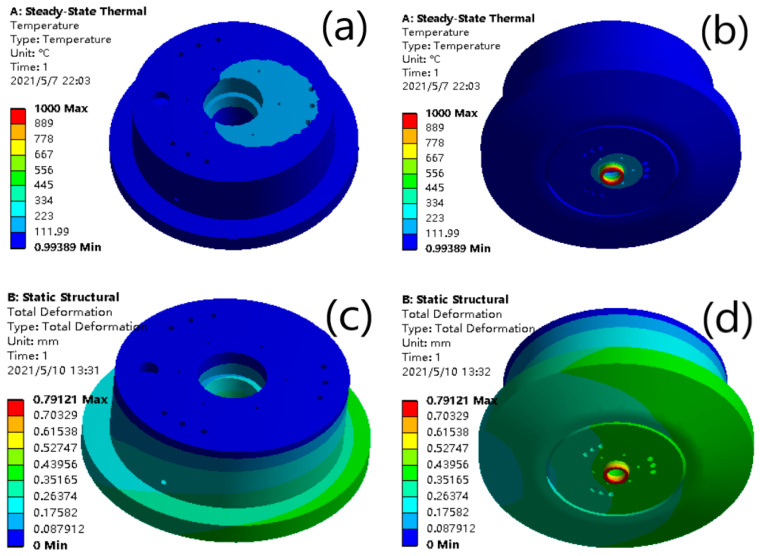
Simulations on the effect of water cooling. (**a**) Vertical view of steady-state thermal; (**b**) bottom view of steady-state thermal; (**c**) vertical view of total deformation; (**d**) bottom view of total deformation; (**e**) vertical view of equivalent stress; (**f**) bottom view of equivalent stress; (**g**) vertical view of equivalent elastic strain; and (**h**) bottom view of equivalent elastic stress.

**Figure 6 materials-14-05196-f006:**
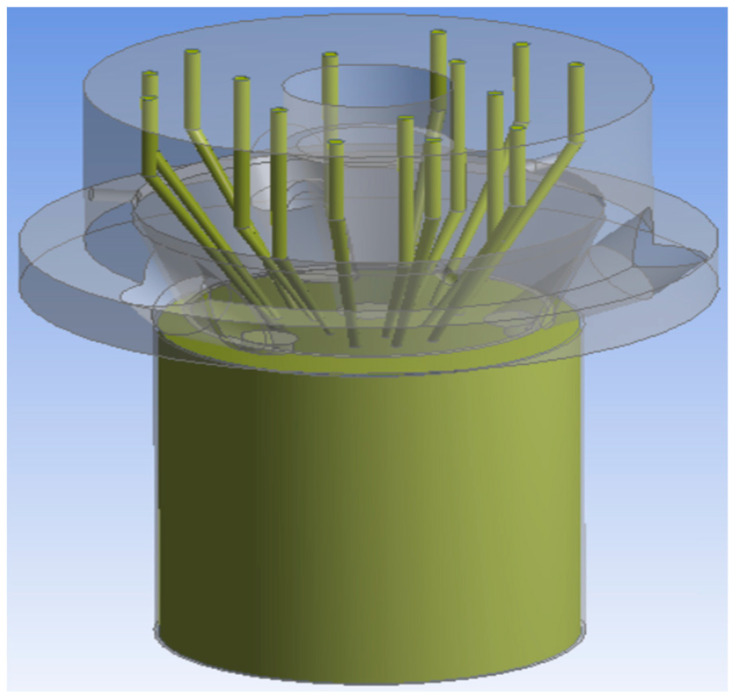
Simulation model.

**Figure 7 materials-14-05196-f007:**
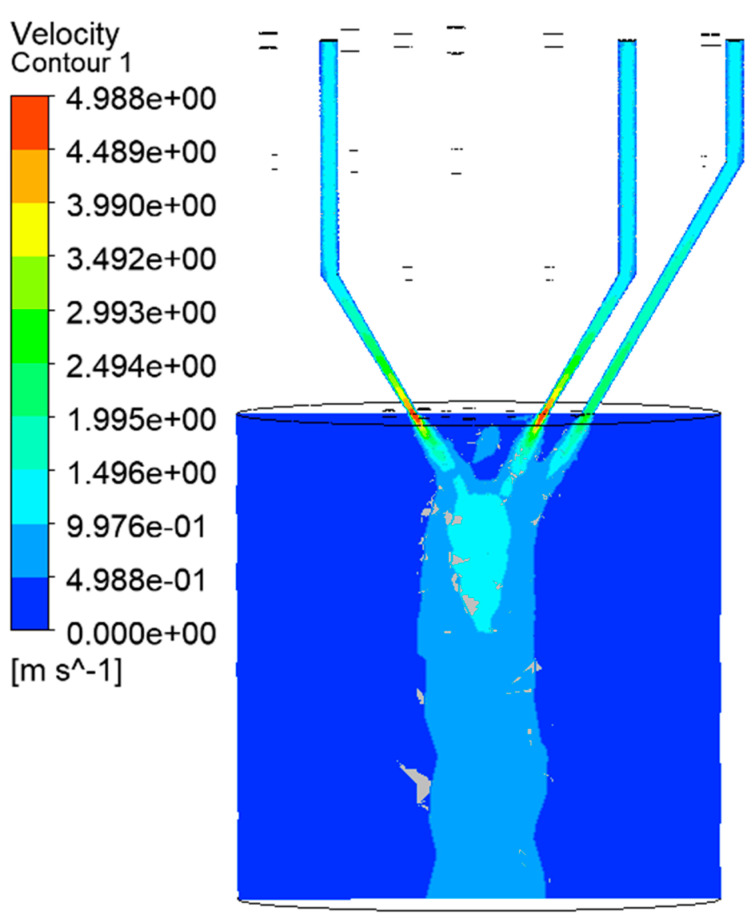
Powder flow velocity diagram of inner and outer rings coordination work.

**Figure 8 materials-14-05196-f008:**
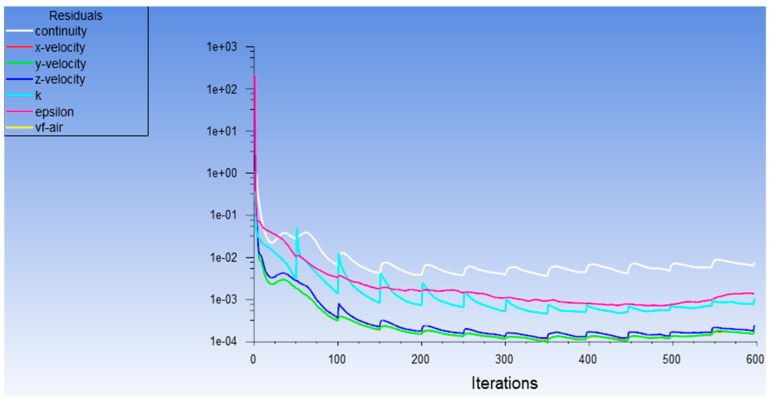
Analysis curve.

**Figure 9 materials-14-05196-f009:**
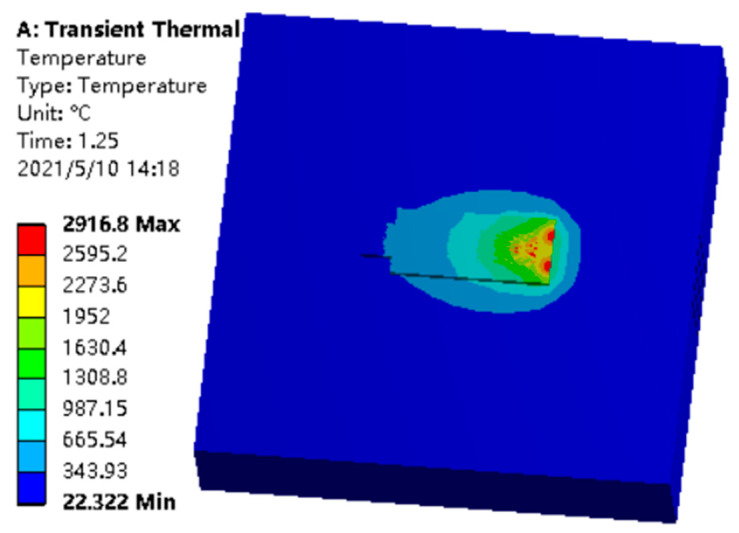
Transient thermal simulation of multiple laser beams working together.

**Figure 10 materials-14-05196-f010:**
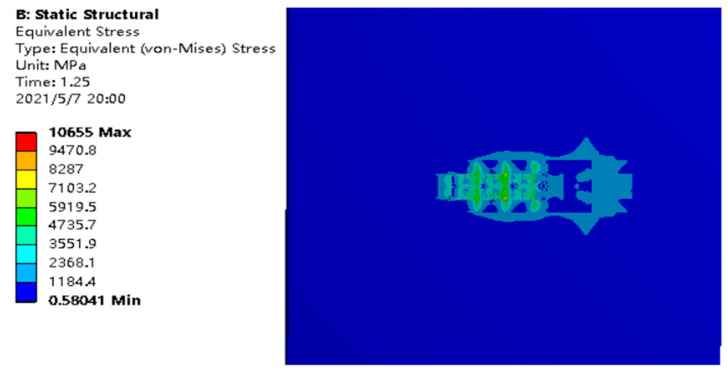
Stress under thermal stress.

**Figure 11 materials-14-05196-f011:**
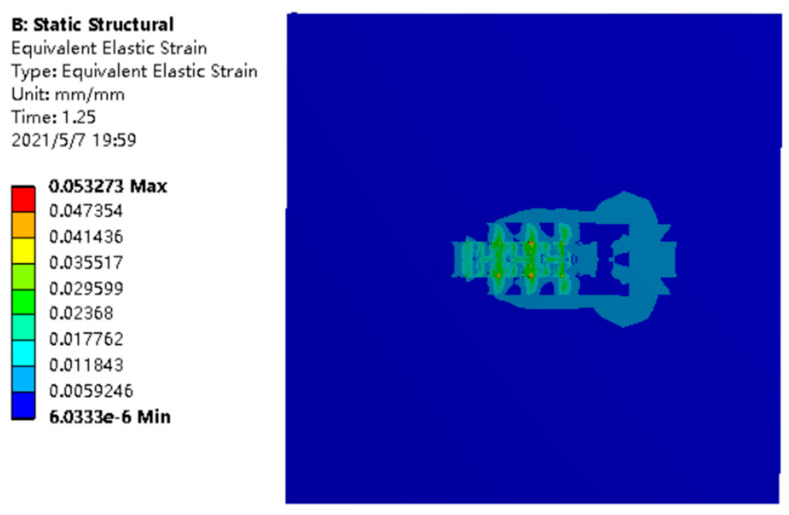
Equivalent strain under thermal stress.

**Table 1 materials-14-05196-t001:** Important parameters and expression formula.

Important Parameters of Continuous Medium	Expression Formula
Density	*ρ* = *m*/*v*
Viscosity; viscosity coefficient	$\tau =\mu \frac{du}{dy}$; $\frac{\mu }{\mu_{0}}=\;{(\frac{T}{T_{0}})}^{n}$
Flow	$Q=\iint_{A}\nu •ndA$
Compressibility	$\beta =\frac{1}{\rho }•\frac{d\rho }{dP}$

**Table 2 materials-14-05196-t002:** Different thermal analysis models and their simulation methods.

Important Parameters of Continuous Medium	Expression Formula
Coupled heat flux model at powder scale	1. Finite volume method (FVM) 2. Lattice Boltzmann method (LBM) 3. Arbitrary Lagrangian Eulerian method (ALE)
Heat flux coupling model based on continuum assumption	FVM Finite element method (FEM)
Heat conduction model based on continuum hypothesis	FEM

## Data Availability

Data available in a publicly accessible repository.

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
