# Peer review of "Design a New Type of Laser Cladding Nozzle and Thermal Fluid Solid Multi-Field Simulation Analysis"

_materials, 2021, doi:10.3390/ma14185196_

Round 1

Reviewer 1 Report

It’s a basic, but interesting research report concerning designing of laser cladding nozzle. The quality of the presented research results and discussions are at acceptable level. However, the following comments/suggestions should be taken into consideration by the authors:

  1. The manuscript should be verified by the authors in terms of spelling (for example "Mpa" or "2000oCin" in conclusions). The font size in equations should be made uniform. Generally it is written carelessly and should be corrected in this regard.
  2. Figure 8 should be enlarged – axis descriptions and the legend are hardly legible.
  3. Figure 5 – the quality of the descriptions of the individual parameters distributions - on the left side of each model - should be improved. They are hardly legible.

Reviewer 2 Report

In this paper, a new kind of cladding nozzle is designed involving six inner powder feeding channels and nine outer powder feeding channels, which would be the proposed novelty.

I have the following comments which should be addressed:

  1. It is not clear why „new heat source” is a keyword. What would be that new source of heat besides the laser?
  2. At the end of introduction, the author must show clearer the necessity of their proposal and justification of doing the research work as presented in the paper, along with a short description of this research.
  3. It is not clear how the simulations were conducted and how the figures 3, 4 and 5 were obtained. What kind of software was used for these simulations? More details are needed.
  4. Since a new kind of cladding nozzle is proposed, the authors should comment on advantages compared to known/old approaches, if any. The authors needs to explain more the difference/advantage of their idea compared to known approaches.
  5. The English style should be improved throughout the paper in order to make it more fluent
